# Utargetome: A targetome prediction tool for modified U1-snRNAs to identify distal-target positions with improved selectivity

Paolo Pigini[1], Federico Manuel Giorgi[2], Keng Boon Wee[1]*

1 Institute of Molecular and Cell Biology (IMCB), Agency for Science, Technology and Research (A* STAR), Republic of Singapore, 2 Department of Pharmacy and Biotechnology, University of Bologna, Bologna, Italy

* weekb@a-star.edu.sg

## Abstract

The endogenous U1 small nuclear RNA (U1-snRNA) plays a crucial role in splicing initiation through base-pairing to donor splice sites (5′-SSs). Likewise, modified U1s that carry a mutation-adapted 5′-terminal sequence have been demonstrated to rescue exon splicing when this is disrupted by genetic mutations within the 5′-SS. Given the base-pairing flexibility of the endogenous U1, the selectivity of modified U1s requires investigation. We developed a computational pipeline (Utargetome) that considers combinations of mismatches and alternative annealing registers to predict the transcriptome-wide binding sites (or targetome) of a U1. The pipeline accuracy was tested by recapitulating well-established alternative annealing registers and specificity for 5′-SSs in the predicted targetome of the human endogenous U1. It was then applied to analyse the targetome of 54 modified U1s that have been demonstrated to restore exon inclusion when affected by 5′-SS pathogenic mutations. While the targetome size was found to be wide-ranging, the off-target load appeared to be reduced for U1s targeting distal sites from the canonical U1-binding position. This feature was predicted also for a large set of 30,204 newly designed U1s targeting 839 5′-SS pathogenic mutations that were expected to affect exon inclusion. Targetome analysis indeed revealed an optimal distal-targeting position at 3 nucleotides downstream from the canonical 5′-SS, for which a modified U1 is likely to have minimal off-targets at 5′-SSs and acceptor splice sites (3′-SSs). Based on these insights, we propose to implement targetome prediction in the design and optimization of therapeutic U1s with improved selectivity.

## Author summary

In the context of evolving gene therapy technologies that demand higher precision and safety, we present Utargetome, a computational tool designed to

---

**Data availability statement:** All relevant data are within the manuscript and its Supporting information files. Source codes for Utargetome are available at available at https://github.com/ppigini/utargetome.

**Funding:** The study was funded by the Agency for Science, Technology & Research, A*STAR (Industry Alignment Fund - Pre-positioning Program H20H6a0027 to KBW) and the Italian Ministry of University and Research (programs PON "Ricerca e Innovazione" 2014–2020, PRIN project 2022CEHEX8, and PNRR program for HPC, Big Data, and Quantum Computing to FMG). The funders had no role in study design, data collection and analysis, decision to publish, or preparation of the manuscript.

**Competing interests:** The authors have declared that no competing interests exist.

predict the binding sites of modified U1-snRNAs across the transcriptome. U1 plays a key role in splicing by binding to specific sites on RNA, and modified U1s have been used to restore normal splicing in cases where splice sites are lost due to mutations. However, ensuring the selectivity of these modified U1s, particularly avoiding unintended off-target effects, is critical for their therapeutic application. Utargetome predicts both the intended (on-target) and unintended (off-target) binding sites of U1s, accounting for mismatches and alternative binding registers. Our findings from analysing more than 30,000 modified U1s show that U1s targeting positions slightly downstream of their typical binding site have fewer off-target events. The insight enables the design of precise U1-based therapies for genetic disorders caused by splicing defects, and towards the advancement of safer gene therapies.

## Introduction

U1-snRNA is an evolutionarily conserved non-coding RNA. Encoded by the *RNU1-1* gene in humans, it is 164 nucleotide (nt)-long, and contains an Sm motif and four stem-loop structures (SLI, SLII, SLIII and SLIV), which interact with multiple proteins to form small nuclear ribonucleoproteins (snRNPs). U1 snRNP initiates exon splicing through base-pairing between 11 nucleotides at its 5′ end (5′-AUACUUACCUG-3′), here called binding sequence, and 5′-SSs [1,2], followed by the recruitment of other snRNPs for the assembly of the spliceosome. Besides splicing, U1 prevents premature transcription cleavage and polyadenylation through binding along the nascent transcript and inhibiting nearby polyadenylation signals, a process known as "telescripting" [1,3]. In addition, a potential role in transcription initiation and directionality was proposed, although the exact mechanism is unclear [1]. The various functions of U1 may account for it being one of the most expressed snRNAs in the cell, with a multitude of known paralogs (over 140 in humans) [1].

An estimated 15% of pathogenic mutations result in mRNA splicing defects [4]. Mutations that occur at 5′-SSs can disrupt endogenous U1 binding with the consequence of aberrant splicing, manifested as exon skipping or intron retention [4–6]. Adaptation of the 5′-terminal sequence of the endogenous U1 to a mutant 5′-SS as a therapeutic strategy for restoring splicing and rescuing wildtype expression has been shown with the use of exogenous U1-snRNAs [4]. The minuscule sequence length of engineered U1s as compared to gene replacement and gene editing approaches confers advantages in manufacturing and for vector delivery. Despite the numerous proof-of-principles in which efficacy of engineered U1s was observed in disease models, including spinal muscular atrophy, cystic fibrosis, hemophilia and neurofibromatosis [4], no U1 to our knowledge has progressed to human study.

Non-selectivity of engineered U1s may indeed be a critical limiting factor, especially given the tolerance for base-pairing mismatches and alternative annealing registers of the endogenous U1 [7–12], whose mRNA-binding sequence is fully complementary to only 0.85% of all 5′-SS sequences [1]. Potential off-target effects

include: 1) promoting the inclusion of alternative or cryptic exons [13–16], which may function as "poison exons" [17] that affect transcript stability; 2) interfering with the activity of other splicing elements, especially in proximity of 3′-SSs [18–21]; 3) inhibiting normal transcript cleavage and polyadenylation at 3′ UTRs [22–24]. Based on a small number of transcriptome-wide studies [25–27], the preliminary conclusion is that engineered U1 off-targets are present, albeit limited. In spinal cord tissues from mice expressing a modified U1, 12 among the 12,414 investigated genes were observed to be up- or down-regulated [25]. The magnitude of differentially expressed genes was similar in liver tissues from mice treated with a modified U1, with expression changes in 13 out of ~13,000 genes and splicing changes in less than 0.1% of transcripts [26]. In human HEK293 cells expressing a modified U1, only one differentially expressed gene and two alternative splicing events were observed [27]. By contrast, an *in silico* study predicted 1,827 perfect matches to the human transcriptome for a modified U1 [28].

As off-targets are dependent on a U1 binding sequence, this warrants an in-depth investigation of their relationship. Utargetome, available at https://github.com/ppigini/utargetome, is a new analysis tool to predict and characterize the transcriptome-wide targets (here referred to as targetome) of a given U1 by considering, besides perfect matches, targets that originate from Watson-Crick base-pairing mismatches and alternative annealing registers, which the endogenous U1 is known to tolerate [7–9]. The targetome was analysed for the total number of targets and their relative position to splice sites. To facilitate assessment between target counts and U1 binding capability, targets were progressively filtered by decreasing the number of minimum annealed bases (MABs), defined as the minimum number of canonical Watson-Crick base-pairings between U1 and target mRNA. Accordingly, 11 MABs is most selective as every base of the 11-nt U1 antisense sequence is paired to the target strand, which can include bulges but no mismatch. Whereas at 9 MABs for instance, there are 11, 10 or 9 canonical base-pairings between U1 and target strand. As a validation, the pipeline was first applied to reproduce the targetome of the human endogenous U1. Thereafter, the targetome of 54 published U1s that have been experimentally validated for their efficacy as therapeutic candidates were investigated and it was found that the sizes of their targetome span several orders of magnitude, indicating a wide range of selectivity. Lastly, 30,204 U1s were newly designed to target 839 5′-SS pathogenic mutations that were predicted to impair exon inclusion. Analyses of their targetome revealed a specific U1 targeting window in proximity of the canonical 5′-SS position that has minimal off-targets at 5′-SS and 3′-SS. This study underscores the need for and the advantage of integrating selectivity as a parameter in U1 engineering. The pipeline developed for U1 targetome prediction could therefore facilitate the discovery of U1 therapeutic candidates with improved selectivity.

## Results

### Survey of potential pathogenic targets for modified U1s

Genetic variants that can potentially be rescued by engineered U1s were surveyed. Unique pathogenic variants localized within the canonical U1 binding site (base positions from -3 to +8 from the exon-intron boundary, Fig 1A) were extracted from the ClinVar database (see "Methods"). Such variants are likely to result in proximal exon skipping, alternative or cryptic donor splice site usage, or adjacent intron retention. The list excluded variants located at base positions +1 or +2, since they would abrogate the entire splicing process [2] and are thus not likely rescuable by engineered U1s. Variants located at predicted mismatched positions between the transcript and the endogenous U1 (Fig 1A) were also excluded, for they are unlikely to diminish U1 binding. A total of 839 potential target mutations were identified in 763 distinct exons encoded by 500 different genes (Fig 1B and S1 Table). This highlights the broad applicability of engineered U1s as a therapeutic strategy, with only a handful of the identified pathogenic variants that have been previously addressed in U1 therapeutic studies (see below).

### Pipeline for transcriptome-wide prediction of U1 targetome

The U1 targetome prediction was based on the Watson-Crick base-pairing rule between the U1 binding sequence and the transcriptome-wide RNAs. Besides targets with perfect complementarity (labelled as "COM"), the predicted targetome includes targets that form alternative annealing registers and base-pairing mismatches (suffixed with "+mm", Fig 2,

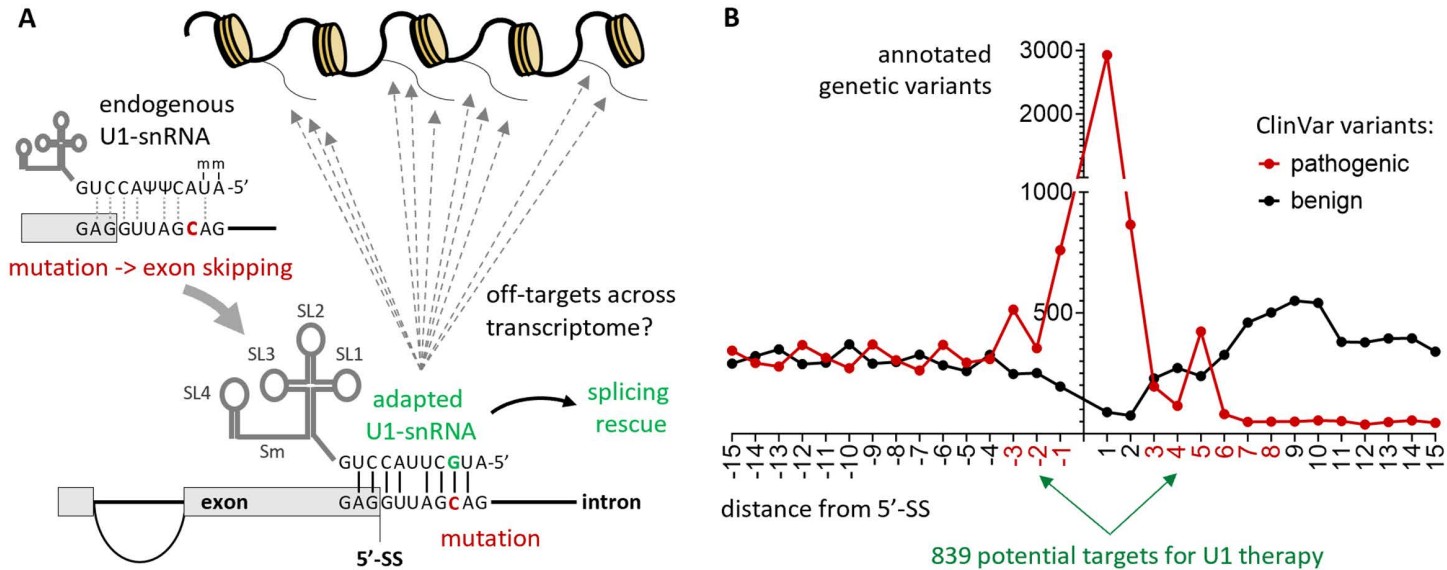

**Fig 1. Design of modified U1s and landscape of 5-SS mutations. (A)** General design of a modified U1 tailored to a specific mutation. The binding sequence is engineered from the endogenous U1 in order to rescue a 5′-SS mutation. The endogenous U1 binding sequence is represented as 5′-mAmUACΨΨACCUG-3′, where "m" represents 2′-O-methylation and "Ψ" is a pseudouridine. Methylation and pseudouridines are not represented in the modified U1, due to lack of prior knowledge. The canonical binding site is represented as 5′-GAGGUUAGCAG-3′. **(B)** Number of annotated "pathogenic" and "benign" mutations extracted from ClinVar and occurring in proximity of 5′-SSs.

module A). Specifically, the annealing registers consist of single- or double-nucleotide bulges on the target RNA, on the U1 or on both strands (labelled as "BS1", "BS2", "BA1" and "BA2"), as well as asymmetric loops on both strands (labelled as "ALS" and "ALA"), which have all been shown to be utilized by the endogenous U1 [7–9]. Base-pairing mismatches are considered in the alternative annealing registers when they occur at least one nucleotide apart from a bulge or loop (Fig 2, module A, and S1 Fig).

The pipeline workflow involves the following sequential steps (Fig 2, see also "Methods"). Given a U1 sequence and the desired MAB, the targetome is built by 1) removing or inserting nucleotides in the target sequence to reproduce loops and bulges and 2) gradually inserting mismatched positions until the number of base-pairs reaches the input MAB (Fig 2, module A, and S1 Fig). Specifically, BS1 and BS2 (single- and double-nucleotide bulges on target strand) are simulated by inserting 1 and 2 nt respectively, BA1 and BA2 (single- and double-nucleotide bulges on the U1 strand) are simulated by deleting 1 and 2 nt respectively, ALS (asymmetric loops with the larger loop on the target strand) are simulated by deleting 1 nt and inserting a new combination of 2 nt, ALA (asymmetric loops with the larger loop on the U1 strand) are simulated by deleting 2 nt and inserting a new nt, and base-pairing mismatches are subsequently implemented as new combinations of nucleotides, with at least one nucleotide apart from the bulge(s)/loop (if present) to preserve their hypothetical structure. As the tolerance for G:U wobble pairing under a wide repertoire of modified U1 sequences is not known, the current implementation treats G:U as mismatches. Target sites with incidence of wobble pairs will thus have lower MABs than when wobble pairs are considered explicitly.

Next, each target sequence in the list is BLASTed against all genomic sequences that produce annotated transcripts, including introns (Fig 2, module B); mitochondrial or chloroplast (for *A. thaliana*, see below) genes are excluded, since U1 activity resides in the nucleus [2]. The hits obtained are subsequently analysed by the following criteria (Fig 2, module C): 1) hits on the same genomic position but mapped to different annotated transcripts are counted as distinct, since they would affect different RNA molecules; 2) hits produced by different annealing registers but mapped to the same position in

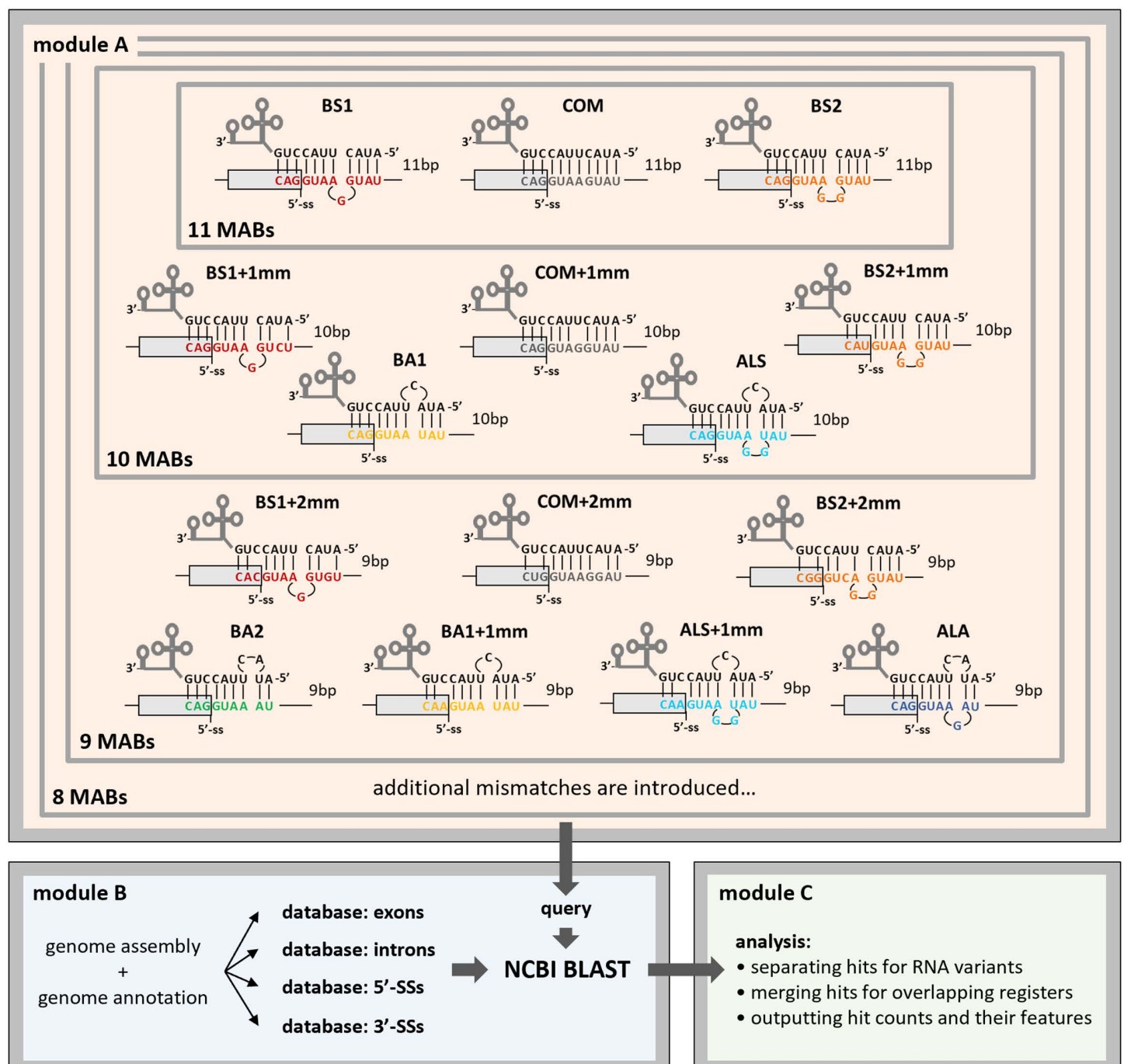

**Fig 2. Pipeline for transcriptome-wide prediction of U1 targetome.** (**Module A**) The potential targetome of the binding sequence (input sequence) of a given U1 was predicted through the consideration of six alternative annealing registers and the presence of mismatches in the input sequence or in the sequences with alternative registers. Annealing registers include: regular annealing with no alternative registers ("COM"), single- and double-nucleotide bulges on the target strand ("BS1" and "BS2", respectively), single- and double-nucleotide bulges on the U1 strand ("BA1" and "BA2", respectively), asymmetric loops with the larger loop on the target strand ("ALS"), asymmetric loops with the larger loop on the U1 strand ("ALA"), and mismatched positions ("mm"). The number of annealed base-pairs (indicated as bp) for each combination of annealing register and mismatches corresponds to the number of vertical lines connecting the annealed strands in module A; representative MABs of 11, 10 and 9 are depicted. (**Module B**) Every possible target sequence generated from the input sequence was BLASTed against a database of exons, introns, 5′-SSs or 3′-SSs. (**Module C**) BLASTed hits obtained in module B were processed as follows: 1) hits on the same genomic location but on different transcript variants were considered

as distinct; 2) hits on the same position of the same transcript variant produced from different registers were merged. The processed hits were classified and broken down by their annealing registers and by their relative position within exons, introns, 5′-SSs or 3′-SSs.

a transcript (i.e., sharing the same 5′-most base position) are counted as a unique hit, since their effect on the transcript is expectedly identical. Finally, the hits are classified by the genomic annotation of their loci into four categories: hits mapped entirely in exons or introns are classified as "exonic" or "intronic targets" respectively; hits overlapping with 5′-SSs or 3′-SSs are classified as "5′-SS" or "3′-SS targets" respectively. The position of a predicted target in reference to a nearby 5′- or 3′-SS always refers to the position of the 5′-most nucleotide on the target sequence, i.e., the 3′-most nucleotide on the U1 antisense sequence, regardless of whether the base-pairing consists of a canonical Watson-Crick or a mismatched pairing (S1 Fig). The sum of all targets from these four categories constitutes the full size of the U1 targetome.

**Targetome prediction of the endogenous U1**

The pipeline was applied to obtain the targetome of the human endogenous U1 (*RNU1-1*) and for mapping and characterizing its transcriptome-wide binding sites as a means for method validation. The 5′-first 11 nucleotides of the *RNU1-1* transcript (5′-AUACUUACCUG-3′) were used as input binding sequence. To recapitulate the flexibility of *RNU1-1* in base-pairing mismatches and alternative annealing registers, the targetome size was analysed at decreasing numbers of annealed bases, down to 6 MABs, for which every target in the targetome has at least six bases annealed to the U1 binding sequence. This ~55% minimum complementarity corresponds to the minimal 14–15 hydrogen bonds required for functional U1 binding [10–12,15]. For comparative controls, targetomes were obtained for the endogenous U1 of two evolutionarily distant eukaryotic species, the plant *Arabidopsis thaliana* and the amoeba *Dictyostelium discoideum*, both of which have identical binding sequence as *RNU1-1* [29]. As a negative control in each species, the complementary sequence of the input sequence, c(RNU1-1), was used.

Both the number of RNU1-1 and c(RNU1-1) targets increases logarithmically by five orders of magnitude in each species when MABs decrease from 11 to 6 (Figs 3A, S2A and S3A). Fig 3B depicts the human RNU1-1 targetome composition by both target locations and annealing registers as a function of MABs. While most targets are localized in introns, which are generally one order of magnitude longer than exons [30], 5′-SS targets disproportionately constitute the next largest fraction from 11 to 9 MABs, suggesting RNU1-1 selectivity for these sites. The same trend was also observed in the respective targetome of *A. thaliana* and *D. discoideum* (S2B and S3B Figs); of note, the fraction of exonic targets is significantly larger than the fraction of intronic targets in both species, which is probably due to the higher representation of exon-coding regions in their genomes. Below 9 MABs in the three species, more targets localize to exons than at both splice sites combined, which is in concordance with the relative proportions of exonic and splice site positions in the genome. With regard to the annealing registers, BS1 and BS2 (single- and double-nucleotide bulges on the target strand, respectively) are the most common (Figs 3B, S2B and S3B, rectangular bars). As the variety of registers increases, they appear to be evenly distributed at the four target locations (S4 Fig). Importantly, the pipeline was able to identify six known 5′-SS targets that each interacts with the endogenous U1 via alternative registers [7,9] (S5 Fig and S2 Table).

Considering that canonical 5′-SS binding sites (located between position -3 and +8 from the exon-intron boundary) constitute the key functional targetome of the endogenous U1, targets located at this position were further analysed. Target counts at canonical 5′-SSs in *H. sapiens*, *A. thaliana* and *D. discoideum* follow a sigmoidal trend with decreasing MABs, and respectively cumulate to 96.7%, 97.6% and 96.5% of the corresponding total annotated 5′-SSs at 6 MABs (Figs 3C, S2C and S3C). By contrast, canonical 5′-SS target counts for c(RNU-1) are not significant in each species. This further corroborates the minimum of 14–15 hydrogen bonds (approximately 6 MABs) required for the functional binding of the endogenous U1 [10–12,15]. In the remaining small percentage of annotated 5′-SSs that were not matched to the canonical 5′-SS targets in the targetome, many of the splice sites carry an alternative dinucleotide instead of the typical

**Fig 3. Targetome of the human endogenous U1.** The 5′-terminal nucleotides of *RNU1-1* transcript (5′-AUACUUACCUG-3′) were used as the input binding sequence for the pipeline. Targetome of its complementary sequence was analysed as control. MABs from 11 to 6 (corresponding to 100% and 55% complementarity respectively) were considered. **(A)** Targetome size for the endogenous U1, labelled as "RNU1-1", and the control, labelled as "c(RNU1-1)", with decreasing MABs. **(B)** Breakdown of the targetome composition of the endogenous U1 by target locations (pies) and annealing registers (rectangular bars). **(C)** Target counts overlapping canonical 5′-SSs (from position -3 to +8 from the exon-intron boundary) for the endogenous U1 and c(RNU1-1) with decreasing MABs. The dashed horizontal line indicates the total number of annotated 5′-SSs in all transcripts and variants as a reference. **(D)** Target distribution in the proximities of canonical 5′-SSs in the targetome of the endogenous U1 (left) and c(RNU1-1) (right) as a function of MABs. Sites are 1 nt apart, ranging from 15 nt up- to 10 nt down-stream of the exon-intron boundary. All positions are referenced to the 5′-most position of the target sequence.

GT at positions +1 and +2 (S6 Fig), and thus possibly suggests an alternative splicing mechanism [31]. As this can also be attributed to the endogenous U1 mediating splicing at non-canonical positions [31,32], targets in the proximity of 5′-SSs, which lie within positions -15 to +10 from the splice site (referred to the 5′-most position of the target sequence from the exon-intron boundary), were interrogated. In addition to the highest enrichment at the canonical position as expected, 5′-SS targets of the endogenous U1, but not c(RNU-1), were enriched consistently at specific distal positions across the three species (Figs 3D, S2D and S3D). Further analysis of the nucleotide compositions at these distal positions revealed a CAG motif among the three species (S7 Fig), which may be an evolutionary conserved binding motif for mediating splicing

at non-canonical sites. Similarly, targets in proximity of 3′-SSs were also found in all three species, with a significant enrichment between positions -3 and +8 from the intron-exon boundary (S8 Fig), which is consistent with previous findings indicating possible direct binding of the endogenous U1 to 3′-SSs [33]. In conclusion, the pipeline is able to capture the essential characteristics of the targetome of the endogenous U1, which shall be the benchmark for the analysis of the targetome of modified U1s.

Lastly, the Gibbs free energy (ΔG) of U1:target duplexes were evaluated for targets of the endogenous human U1 predicted at positions overlapping with 5′-SSs, 3′-SSs or exonic regions at MABs from 11 to 6 [7]. As expected, ΔG is inversely correlated with MABs (S9A Fig). ΔG of duplexes at 5′-SS are generally lower than at the other two positions (S9B Fig), which may suggest U1 preferential binding to 5′-SSs, a well-established fact.

## Targetome analysis of modified U1s

The human targetome of 54 modified U1s was predicted with the pipeline described in the previous paragraph. Their efficacy in restoring exon inclusion, affected by 23 different 5′-SS pathogenic mutations, has been validated in 16 different studies as a potential therapeutic strategy for a variety of diseases (S3 Table). Each U1 carried a binding sequence of 11 bases with no dinucleotide TT, GA, or GG at the 5′-end, which negatively affect its stability [34] and could therefore influence its on- or off-target activity. In all the studies, the modified U1s were expressed from a plasmid vector containing the promoter, scaffold and terminator sequences of human *RNU1-1*. Selectivity of each U1 was inferred from both its full targetome size and number of targets at splice sites, which mediate the default mechanism of action of engineered U1s [13–16,18–21]. The full targetome size across the U1s was found to span four orders of magnitude at perfect complementary, from 15 to 15,817 targets, and from 2,314,404–54,757,018 targets at 9 MABs, which are dominated by intronic targets (Fig 4A and S3 Table). For targets overlapping 5′-SSs, 0–737 perfect hits or 5,202–507,381 hits at 9 MABs were predicted across the modified U1s (Fig 4B and S3 Table). By comparison, the *RNU1-1* targetome size lies approximately at the median of the modified U1s, and it has more 5′-SS targets than 51 of the modified U1s. The latter trend is consistent when targets at distal positions from the canonical 5′-SSs (from 10 nt up- to 25 nt downstream of the exon-intron boundary) [35–37] were included, since their contribution is not significant (S10 Fig). There are significantly fewer targets overlapping 3′-SSs than 5′-SSs, with 0–109 at perfect complementary, and 2,365–71,121 at 9 MABs (Fig 4C and S3 Table).

The pattern of target counts among the 54 U1s is generally similar between perfect complementarity and 9 MABs (Fig 4A, 4B and 4C). The trend is still broadly conserved from 11 to 7 MABs among four representative U1s, each with a unique targetome characteristic (S11 Fig and S4 Table), namely U1-1 (smallest targetome), U1-36 (highest 5′-SS target count), U1-49 (highest 3′-SS target count), and U1-54 (largest targetome). Another observation is that while modified U1s with small targetome size usually have low target counts at both splice sites, this is not always the case. For example, U1-52 possesses the second largest targetome size but has one of the lowest number of 5′-SS targets, whereas U1-36 has the most 5′-SSs targets although its targetome size is the 19th largest (Fig 4A and 4B). Given the essential role of 5′-SSs in mediating U1 function, U1-52 is expected to be more selective than U1-36. Hence, targetome size is not a definite indicator of selectivity.

In order to show that predicted targets are biologically relevant, the 5′-SS targets of U1-36 were further interrogated (Fig 4D and S5 Table). As U1-36 was validated to rescue the skipping of *RHO* exon 4 when affected by c.936G>A mutation in retinopathies [38], RNAseq datasets originating from human retinas were analysed to identify the most probable off-target exons. These exons, besides containing a binding site for U1-36 at their 5′-SS, were selected for having an average percent-splice-in (PSI) of less than 0.5, i.e., they are spliced-out in more than 50% of the coding transcripts (see Methods). Given the relatively low inclusion levels of these exons, a significant increase in PSI induced by U1-36 is likely to have biological implications. A total of 39 candidate exons were identified from 737 perfectly complementary 5′-SS targets (Fig 4E and S6 Table) on which U1-36 can potentially increase their PSI and affect the transcript stability

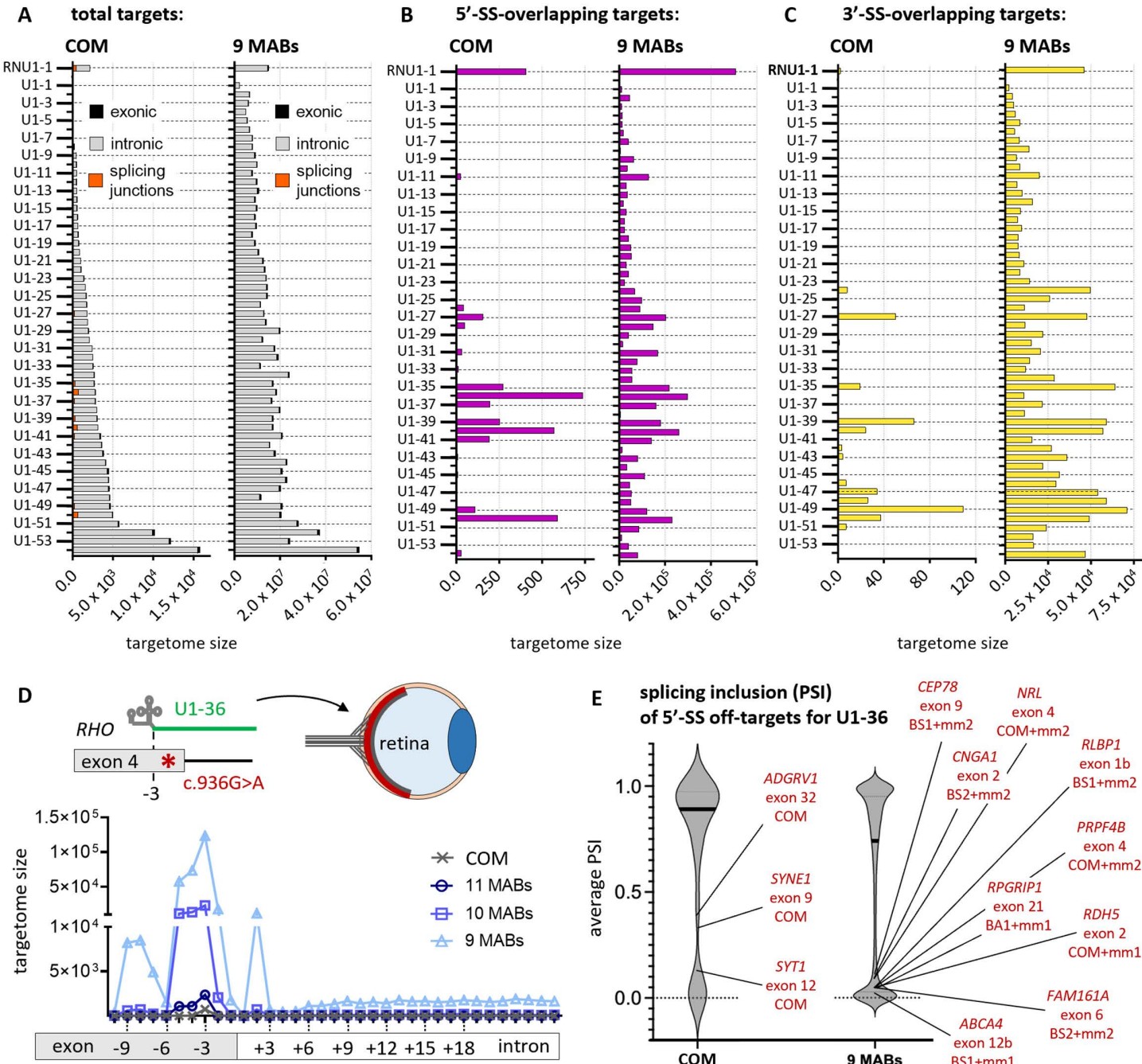

**Fig 4. Targetome analysis of the 54 modified U1s targeting human transcripts.** The targetome of every modified U1 was predicted by the pipeline for the register with perfect complementarity (COM) or for all registers with 9 MABs. **(A)** Targetome size of every modified U1 (refer to S3 Table for the description). Total target counts are broken down by their location within exons or introns or overlapping with splicing junctions. Target counts at **(B)** 5′-SSs and **(C)** 3′-SSs, which include all positions overlapping with the respective splice junction. **(D)** Positional distribution of 5′-SS target counts of U1-36 (S3 Table) from the canonical 5′-SS. Position intervals are 1 nt apart, ranging from 10 nt up- to 25 nt down-stream of the exon-intron junction (with reference to the 5′-most position of the target sequence). The analysis was performed for the register with perfect complementarity (COM) and for all annealing registers from 11 to 9 MABs. **(E)** Average percent-splice-in (PSI) for exons expressed in human retina whose 5′-SS was found in the targetome of U1-36. All targets, with perfect complementarity or 9 MABs, were located from 10 nt up- to 25 nt down-stream of the exon-intron junction. Average PSI values were calculated from eight different RNA-seq datasets derived from human eye retinas. The annealing register with mismatch (if any) for each targets are provided.

or function. Three exons in particular, *ADGRV1* exon 32, *SYNE1* exon 9 and *SYT1* exon 12, are associated with important retina functions [39,40]; in comparison, RNU1-1 is predicted to bind to these exons at non-perfect complementarity with no more than 10 annealed bases (~90% complementarity). As anticipated, the number of candidate exons grows exponentially with decreasing complementarity, with 46,217 candidate exons found from 297,962 5′-SS targets at 9 MABs (Fig 4E and S7 Table). The sheer number of biologically relevant exons predicted from the targetome inevitably increases the probability of actual off-target events. These include exons from several genes with critical roles in retinal biology, such as *ABCA4* [41], *CEP78* [42], *CNGA1* [43], *FAM161A* [44], *NRL* [45], *PRPF4B* [46], *RDH5* [47], *RLBP1* [48], and *RPGRIP1* [49]. Among them, BS1 and BS2 are most common annealing registers which is similar to the endogenous U1 at 9 MABs (Fig 3B). Utargetome can thus be useful to direct experimental efforts in the evaluation of off-targets for a modified U1.

Lastly, it is important to note that modified U1s addressing the same genetic mutation but carrying different binding sequences have distinct targetomes (S12 Fig). An exemplary case is U1-30, which addresses the same mutation as U1-36, but which has one of the least 5′-SSs targets among the 54 U1s (Fig 4A and 4B). Moreover, total targetome size is not an accurate proxy for selectivity – for instance, U1-52, despite having one of the largest targetomes has a relatively small set of 5′-SS targets. In summary, the results suggested that modified U1s can be classified based on their selectivity, and is highly (binding) sequence-dependent, indicating that the design of the U1 binding sequence can be leveraged to minimise off-targets, as shown below.

### Distal targeting strategy to mitigate U1 off-targeting

Considering that sequences are generally less conserved at distal positions than at canonical 5′-SSs, the well-established mechanism of U1 distal targeting [37] was investigated as a strategy to design U1s with reduced off-targets. This idea is supported by the targetome analysis of distal-targeting U1s amongst the 54 modified U1s (S8 Table). The first case study involves three U1s that rescue the effect of c.9726+5G>A mutation in *F7* exon 9 [35,50]. Their respective target positions, with reference to the 5′-most position of the target sequence, are +17 (U1-8), -10 (U1-14) and -3 (U1-50). With both perfect complementarity and 9 MABs, distal-targeting U1-8 and U1-14 have substantially smaller targetome size and target counts at both 5′-SSs and 3′-SSs than the canonical-targeting U1-50 (Fig 5A and S8 Table). The second case study considered three distal-targeting U1s rescuing the effect of c.669A>T mutation in *F8* exon 6, with respective target positions at +1 (U1-29), +7 (U1-38) and +16 (U1-52) [51]. Although all three U1s showed no 5′-SS targets at perfect complementarity, U1-38 is most selective when considering both the targetome size and 5′-SS targets with 9 MABs (Fig 5A and S8 Table).

With the aim of testing the distal targeting strategy on additional mutations to identify possible optimal distal positions, *de-novo* U1s were designed for the 839 unique 5′-SS mutations that were identified previously in this study (Fig 1B and S1 Table). Specifically, 35 U1s were designed for every mutation by walking their 11 nt-long binding sequence, at one nucleotide resolution, from 10 nt upstream to 25 nt downstream of the exon-intron boundary (with reference to the 5′-most position of the target sequence, Fig 5B and S9 Table), which defines an optimal range for mediating exon rescue [35,36]. A U1 carrying the binding sequence of RNU1-1 but with a single-nucleotide adaptation to each mutation was also included, as it can be effective in some cases (e.g., U1-27, U1-40 and U1-48 in S3 Table). The targetome of each of the 30,204 newly-designed U1s was subsequently predicted by the pipeline at perfect complementarity. Fig 5C depicts the target counts at exons, introns, 5′-SSs and 3′-SSs for each U1 targeting each walking position (also S9 and S10 Tables). An optimal distal position was observed at 3 nt downstream from the canonical 5′-SS or equivalently, at position +1, given that both 5′-SS and 3-SS targets are the lowest for U1s targeting at this position, showing both the lowest median and the lowest count for the U1 with the highest number of targets (S10 Table); p-values<0.01 when counts at this position are compared to counts for U1s targeting most of the other considered positions in all target categories (S10 Table). This was further confirmed when all targets with 10 MABs were considered for position ±1 and the two nearest positions (S13 Fig).

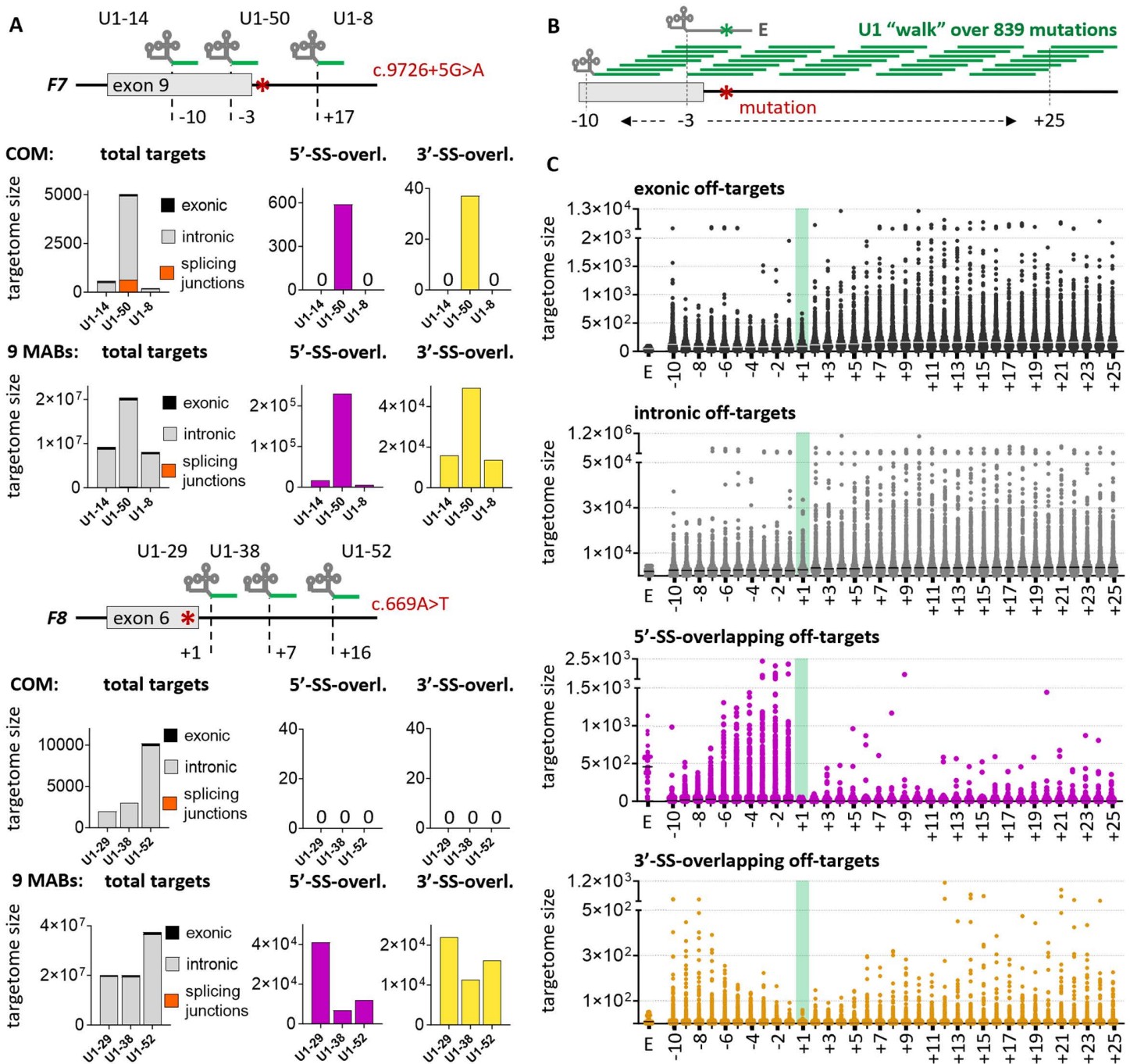

**Fig 5. Targetome analysis of distal-targeting modified U1s. (A)** Target counts of selected U1s that have previously been demonstrated to efficiently rescue exon skipping resulting from mutations *F7* c.9726+5G>A (top) or *F8* c.669A>T (bottom). Except for U1-50, they have been each designed to target distal positions from the canonical 5′-SS. Positions indicate the 5′-most nucleotide of the U1 target sequence, where the canonical site corresponds to "-3". The total targetome size (broken down by their location within exons or introns or overlapping with splicing junctions) and targets overlapping 5′-SSs, or 3′-SSs were predicted for perfectly complementary targets (COM) or with 9 MABs. **(B)** De-novo design of 30,204 U1s targeting 839 unique 5′-SS mutations from ClinVar database that were predicted to affect splicing. The U1s were "walked" along their target transcripts, from 10 nt up- to 25 nt downstream of the exon-intron junction (with reference to the 5′-most position of the target sequence). The endogenous U1 carrying a single-nucleotide adaptation to the mutation (labelled as "E") was also designed. **(C)** Targetome analysis of the 30,204 newly designed U1s at each target position. Counts of perfectly complementary targets are depicted for exonic, intronic, as well as 5′-SSs and 3′-SSs overlapping targets. An optimal target position is located 1 nt downstream of the exon-intron junction (highlighted in green).

On the other hand, the lowest exonic and intronic targets are observed for U1s targeting positions -9, -8, -4, and +1 (S10 Table), which further affirm position +1 as an optimal target position.

Finally, the 23 5′-SS mutations targeted by the 54 published U1s (S3 Table) were revisited, for which optimized U1s targeting position +1 were designed and their targetome predicted at both perfect complementarity and 9 MABs. Consistently, the optimized U1s have significantly lower target counts at both 5′-SSs and 3′-SSs than the 54 published U1s, while no considerable differences in their exonic and intronic target counts were discerned (S14 Fig). In conclusion, the distal targeting strategy is demonstrated to improve the selectivity of modified U1s through reducing off-targeting events, especially at the splice sites, even when allowing mismatches and without the cost of increased off-targets at exons or introns. This forms the basis for the rational design of modified U1s that can be facilitated by Utargetome.

## Discussion

We developed Utargetome, a computational pipeline for predicting the transcriptome-wide binding sites, or targetome, of a U1. In addition to base-pairing mismatches between the U1 binding sequence and target RNAs, the pipeline considers six known alternative annealing registers, which include single- and double-nucleotide bulges and asymmetric loops between the base-pairing strands. Also, the pipeline annotates the relative location of the predicted targets, whether they are exonic, intronic, or overlapping with 5′-SSs or 3′-SSs. The pipeline was first tested to predict the targetome of the human U1, *RNU1-1*, and of both *Arabidopsis thaliana* and *Dictyostelium discoideum*, which all share the same U1 binding sequence. Once validated, the pipeline was applied to obtain and analyse the human targetome of 54 modified U1s, which had been experimentally validated to be effective in restoring exon inclusion as a potential therapeutic strategy for 5′-SS pathogenic mutations. The selectivity of the U1s, which was inferred from both their full targetome size and targets at splice sites, was found to be wide-ranging. However, six modified U1s that were designed to bind distal positions were observed to have significantly reduced off-target counts at both 5′-SSs and 3′-SSs, suggesting that distal targeting can improve U1 selectivity. This evidence was then leveraged for the design of 30,204 U1s targeting 839 unique 5′-SS pathogenic mutations, which collectively implicate the splicing of 763 exons encoded in 500 genes. Analysis of the 30,204 targetomes predicted by the pipeline converge to an optimal U1 distal-targeting position at 3 nt downstream from the canonical 5′-SS, which leads to minimised off-target events, especially at 5′-SSs and 3′-SSs. The results justify the rationale of considering this particular distal position in the design of a U1 binding sequence.

Comprehensive combinations of mismatches and annealing registers were considered by the pipeline as the base-pairing mechanisms between a U1 binding sequence and its RNA binding site. Detailed analysis of the targetome predicted for the endogenous human U1 recapitulated not only the well-established specificity for 5′-SSs and alternative annealing registers, but also less studied aspects such as the enrichment of target sites at 3′-SSs and non-canonical 5′-SSs positions. Future transcriptome-wide RNA–RNA interaction data using psoralen-based crosslinking techniques [33] will be required to ascertain whether these target sites are indeed sites of base-pairing interactions with U1. Nonetheless, a very small fraction of annotated 5′-SSs in the human transcriptome was not found in the predicted targetome. This is inferred to be due to mechanisms not considered by the current pipeline, such as: 1) unelucidated alternative annealing registers; 2) alternative splicing mechanisms for 5′-SSs that harbour atypical dinucleotides at positions +1 and +2 [31]; 3) intron processing through the minor spliceosome, which uses different snRNAs and a distinct 5′-SS motif [31,33]; and 4) additional protein factors possibly mediating U1 binding to RNA targets [2]. Although the predicted targetome for the human endogenous U1 is likely underestimated, it may not necessarily be the case for modified U1s, as they may not utilize such alternative base-pairing mechanisms.

Analysis of the human targetome of 54 modified U1s showed a wide range of off-target counts, spanning a few orders of magnitude in some cases. This facilitates, for the first time, the classification of U1s based on their selectivity and therefore provides a metric for selectivity in the design of the U1 binding sequence. Of relevance, the full targetome size of a modified U1 may have an indirect effect on its efficiency due to a "sponge" effect, in which the on-target site needs to

compete with the transcriptome-wide off-target sites for binding. At the same time, the full targetome size cannot be a definite indicator of selectivity, as it does not always correlate with the number of target sites at both the 5′-SSs and 3′-SSs. Based on the essential role of 5′-SSs in mediating both the on- and off-target functions of a modified U1, the target count therein was consequently chosen as a selectivity indicator, also considering that the major off-target effect would be the increased inclusion levels of alternatively or poorly spliced exons. In support of this, such biologically relevant off-target exons were identified in the targetome of the least selective U1, U1-36. Of note, the pattern of selectivity across different U1s seemed to be generally preserved even when considering alternative registers and mismatches.

The distal-targeting strategy was inspired from the superior selectivity of specific U1s, amongst the published 54 modified U1s, that were designed to bind distal positions. This was corroborated from the targetome of 30,204 newly designed U1s targeting 839 unique 5′-SS pathogenic mutations at distal positions. For these, an optimal target site at position +1 was identified to be associated with the lowest off-target events at both 5′-SSs and 3′-SSs. Of note, novel U1 design at such position also improved the selectivity of most of the 54 modified U1s that were initially evaluated from literature. However, as the binding sequence of a U1 influences its efficiency and selectivity simultaneously, and depending on the specific mutation to rescue, positions outside of +1 may need to be screened to identify optimal U1 candidates. Furthermore, in the presence of proximal splice sites, it is possible that distal targeting may result in the use one of the inactive putative sites [52], again suggesting that each scenario should be evaluated individually.

Utargetome is showed to recapitulate with fidelity major known features of the human endogenous U1. Because of the infidelity of U1 binding, it is tricky to rank or discriminate target sites for their likelihood of binding by either MABs or Gibbs free energy of U1:target duplexes. Nonetheless future experimental validation shall be useful to quantitate the actual off-target rates as a direct comparison with RNAseq–based off-target analyses remains limited by the current literature. The few available transcriptome-wide studies of engineered U1s have reported relatively modest gene expression or splicing changes [25–27]. Furthermore, most of these studies were conducted in mouse tissues. Although one study performed RNAseq in human cells, it did not provide a comprehensive list of off-target events for direct comparison. At the same time, non-sequence based factors such as chemical modifications of the binding sequence (including pseudouridines and methylation) [1], and splicing factors or spliceosome components (such as U1-C, U5 and U6) [2] could be considered for implementing the methodology. In endogenous U1s, two conserved pseudouridines at positions 5 and 6 in the 5′ end are known to enhance the thermodynamic stability of base-pairing with 5′-SSs and contribute to accurate splice site recognition and spliceosome assembly [53–55]. It is plausible that co-transcriptional pseudouridylation may occur on modified U1s as they are being expressed from plasmids or viral vectors [56]. {R1Q1} Analogously, the role of both U5 and U6 snRNAs is likely required for the efficacy of modified U1s including distal-targeting ones since during canonical splicing, they displace U1 and U2 snRNPs from the splice site, stabilize the spliceosomal complex and trigger the ligation of the two exons and intron removal [1,2]. U5 interacts with the last nucleotides of the upstream exon while U6 base-pairs with the intronic side of the 5′ splice site. Future studies are thus required to clarify their mechanistic action and also investigate pseudouridylation on modified U1s and the effect on binding kinetics.

Overall, Utargetome can be readily applied to prioritise or drop out U1s prior to a screen by comparing the selectivity of their predicted targetomes. Moreover, predicted off-targets with biological relevance can inform and guide the analysis of RNAseq data from U1-treated disease models. In conclusion, we propose the application of targetome prediction in the design of U1 binding sequences and expedite the discovery and validation of optimal U1s.

## Methods

### ClinVar analysis

The entire ClinVar database was downloaded (release version in S11 Table), and annotated variants were analysed with Python3.9. Only "pathogenic" variants associated with degenerative disorders were considered, whereas variants labelled as "likely pathogenic", "likely benign", or lacking information about the related pathology and/or their genetic coordinates,

as well as variants related to "cancer", were filtered out. The relative positions of the variants from their respective nearest annotated 5′-SS were determined. Only variants located within the canonical binding site of the endogenous U1, or equivalently from positions -3 to +8 from the exon-intron boundary, were selected. Variants located at positions +1 or +2 were further excluded as they are generally not rescuable by U1 approach [2]. The final selection of variants was based on whether they potentially affect the binding of the endogenous U1. The script and its manual are available for download in GitHub repository, under the name "uvariants" (link in S11 Table).

**U1 targetome prediction pipeline**

The targetome prediction pipeline was coded in Python3.9 under Linux and MacOSX environments and integrated with the BLASTn algorithm [24,57,58] (version in S11 Table). The implementation steps are summarized as follows. First, the input list of U1 binding sequences is converted to target sequences by complementarity. Second, annealing registers (Figs 2 module A and S1) are implemented by deleting or inserting new nucleotide combinations at every possible position, except for the first and last positions: BS1 and BS2 (single- and double-nucleotide bulges on the RNA target strand) are implemented by inserting 1 and 2 nt respectively; BA1 and BA2 (single- and double-nucleotide bulges on the U1 strand) are implemented by deleting 1 and 2 nt respectively; ALS (asymmetric loops with the larger loop on the target RNA strand) are implemented by deleting 1 nt and inserting a new combination of 2 nt; ALA (asymmetric loops with the larger loop on the U1 strand) are implemented by deleting 2 nt and inserting 1 new nt. Base-pairing mismatches between the two strands are subsequently implemented as new combinations of nucleotides, with at least one nucleotide apart from the bulge(s)/loop (if present) to preserve their hypothetical structure. Collectively, the full set of target sequences generated constitutes all possible targets of the given U1. Third, BLASTn is used to search every query sequence from the full set of target sequences generated above, in one or more databases. Each database is built with BLASTn (command `makeblastdb`) from a list of exons, introns, 5′-SSs and 3′-SSs. These sequences are extracted by the pipeline from the annotation and assembly files of a given species (*H. sapiens*, *D. discoideum* or *A. thaliana* in this study, release versions in S11 Table). Mitochondrial genes (and chloroplast genes in *A. thaliana*) are excluded from the databases. Targets sequences are then searched using BLASTn (command `blastn`) with the following settings: `-task blastn-short`; `-word_size` as the exact length of the input sequence (depending on the annealing register, see Figs 2 and S1 for examples); `-strand plus`; `-evalue 1000000000`; `-max_target_seqs 1000000000`. Fourth, BLASTed hits are filtered based on the following criteria: 1) hits on different annotated transcripts but corresponding to the same genomic sequence are counted as distinct, as they might induce different effects; 2) hits produced by different annealing registers but located in the same position of the same transcript (i.e., sharing the same 5′-most position) are counted as single hits. Finally, filtered targets are counted and classified as either exonic or intronic (i.e., fully residing within the exon or intron), or overlapping with 5′-SSs or 3′-SSs. 5′-SS and 3′-SS targets are additionally examined within custom ranges or specific positions from the splice site. The position of a predicted target in reference to a nearby 5′- or 3′-SS always refers to the position of the 5′-most nucleotide on the target sequence, i.e., the 3′-most nucleotide on the U1 antisense sequence, regardless of whether the base-pairing consists of a canonical Watson-Crick or a mismatched pairing (S1 Fig). The command-line version of the pipeline and its manual are available for download in GitHub repository, under the name "utargetome" (link in S11 Table).

**Gibbs free energy (ΔG) of U1:target duplexes**

The ΔG of RNA:RNA duplex formation between a given U1 sequence and its predicted target site were estimated using RNAcofold (version 2.7.0) from the ViennaRNA Package [59]. Sequences were aligned in 5′–3′ orientation and input in the standard format. Only base-pairing interactions between U1 and the target were considered. All predictions were performed using default thermodynamic parameters and temperature settings (37°C). The output ΔG values were directly used to assess the relationship between the number of predicted base pairs and the strength of RNA-RNA binding.

## U1 sequences

The binding sequences of the 54 literature-validated U1s were gathered from each reference (S3 Table). The promoter, scaffold and terminator sequences of the U1 cassette were used to deduce the start and end sites of the binding sequences [60]. Novel U1s were designed for the ClinVar dataset of mutations either by extracting the complementary sequence of the target site from the mutated exon sequences, or by adapting the endogenous U1 sequence to the mutation (S9 Table).

## RNAseq analysis

The following RNAseq datasets of human retinas were downloaded from the SRA repository (S11 Table): SRR15431770, SRR15431758, SRR15351389, SRR15351390, ERR5236661, SRR17467505, SRR15539412, SRR16846779 [61–64]. They were converted to FASTQ format using the SRA toolkit and aligned with the HISAT2 program [65] using the in-built human genome index (build GRCh38, index version 2.0.2-beta). The aligned SAM files were converted to BAM format and then sorted using the SAMtools package [66]. TPM values were calculated with StringTie [67] using the human genome annotation (release version in S11 Table). Total gene expression was calculated as the sum of the TPM values of all transcript variants. Transcripts with TPM < 0.5 and novel transcripts were excluded from the analysis. PSI values for a given exon were calculated as the ratio between the sum of the TPM values of all transcript variants carrying the given exon and the sum of the TPM values of all transcript variants of the same gene. Pathway and ontology enrichment analyses were performed with DAVID [68] (S11 Table).

## Bioinformatics and statistical analysis

The total number of 5′-SSs in the genomes of *H. sapiens*, *D. discoideum* and *A. thaliana* were calculated from the respective annotation files (release versions in S11 Table) using Python3.9. All annotated transcript variants were counted, with same 5′-SSs from different variants constituting separate counts, consistently with the targetome prediction pipeline. Positional Weight Matrices (PWM) for S7 Fig were generated from the U1 target sequences of interest by WebLogo web tool (S11 Table). Statistical analysis was performed with GraphPad Prism 9. For the free energy calculation of RNA-RNA interactions (S9B Fig), simple linear regression was performed with 95% confidence intervals, and the Pearson correlation coefficient was also calculated using 95% confidence intervals. For the targetome size of the 30,204 de-novo designed U1s (Figs 5C and S13 Fig), statistical P-values were calculated with one-way ANOVA for repeated values using Tukey's correction, assuming Gaussian distribution of residuals, and performing multiple comparisons between the mean off-target count of each target position and the mean of every other position. For the targetome size of U1s with optimized design (S14 Fig), statistical P-values were calculated with paired t-test and corrected for False Discovery Rate using the Benjamini-Hochberg method.

## Supporting information

**S1 Fig. Representation of targetome creation for a given U1 in the Utargetome pipeline.**
(DOCX)

**S2 Fig. Targetome of the endogenous U1 of *A. thaliana*.**
(DOCX)

**S3 Fig. Targetome of the endogenous U1 of *D. discoideum*.**
(DOCX)

**S4 Fig. Proportion of alternative annealing registers in the targetomes of the endogenous U1 in *H. sapiens*, *A. thaliana* and *D. discoideum*.**
(DOCX)

**S5 Fig.  Reported donor splice sites bound by the endogenous U1 through alternative annealing registers.**
(DOCX)

**S6 Fig.  Dinucleotide combinations at positions +1 and +2 of 5′-SS targets of the endogenous U1 in *H. sapiens*, *A. thaliana* and *D. discoideum*.**
(DOCX)

**S7 Fig.  Potential binding motifs of the endogenous U1 at distal positions in *H. sapiens*, *A. thaliana* and *D. discoideum*.**
(DOCX)

**S8 Fig.  Target distribution in proximity of 3′-SSs for the endogenous U1 and c(RNU1-1) in *H. sapiens*, *A. thaliana* and *D. discoideum*.**
(DOCX)

**S9 Fig.  Analysis of free energy of binding for the predicted targets of the human endogenous U1 at positions overlapping with 5′-SSs, 3′-SSs and exonic regions.**
(DOCX)

**S10 Fig.  5′-SS targets for the 54 modified U1s.**
(DOCX)

**S11 Fig.  Correlation between targetome size and decreasing complementarity for U1-1, U1-36, U1-49 and U1-54.**
(DOCX)

**S12 Fig.  Targetome of the 54 modified U1s grouped by target mutation.** Refer to Fig 4A–4C for the legend.
(DOCX)

**S13 Fig.  Targetome analysis at 10 MABs for de-novo designed U1s targeting 839 unique 5′-SS mutations at selected distal positions (-1, +1 and +2).**
(DOCX)

**S14 Fig.  Comparison of targetome size between modified U1s validated in literature and the newly designed U1s targeting the distal position +1.**
(DOCX)

**S1 Table.  Potential ClinVar pathogenic variants amenable for U1 therapy.**
(XLSX)

**S2 Table.  Predicted 5′-SS targets (canonical position) with 9 MABs for the human endogenous U1 snRNA.**
(XLSX)

**S3 Table.  Predicted targetome for the 54 modified U1s validated in literature.**
(XLSX)

**S4 Table.  Predicted targetome for 4 selected modified U1s validated in the literature (U1-1, U1-36, U1-49 and U1-54).**
(XLSX)

**S5 Table.  Targetome of U1-36 ("RHO_4_-1G>A_1").**
(XLSX)

**S6 Table. Average PSI of exons expressed in human retina whose 5′-SS was found in the targetome of U1-36 with perfect complementarity.**
(XLSX)

**S7 Table. Average PSI of exons expressed in human retina whose 5′-SS was found in the targetome of U1-36 with 9 MABs.**
(XLSX)

**S8 Table. Targetome of literature-validated modified U1s targeting distal positions.**
(XLSX)

**S9 Table. Targetome of 30,204 newly designed U1s targeting distal positions.**
(XLSX)

**S10 Table. Median and maximum target counts and statistical significance of 30,204 newly designed U1s.**
(XLSX)

**S11 Table. Web resources.**
(XLSX)

## Acknowledgments

We acknowledge Prof. Francesc Xavier Roca Castella for inspiring this work, and Wei Yuan Cher for advising the development of the targetome prediction pipeline.

## Author contributions

**Conceptualization:** Paolo Pigini, Federico Manuel Giorgi, Keng Boon Wee.

**Data curation:** Paolo Pigini.

**Formal analysis:** Paolo Pigini.

**Funding acquisition:** Keng Boon Wee.

**Investigation:** Paolo Pigini, Keng Boon Wee.

**Methodology:** Paolo Pigini, Federico Manuel Giorgi, Keng Boon Wee.

**Project administration:** Paolo Pigini, Keng Boon Wee.

**Resources:** Paolo Pigini.

**Software:** Paolo Pigini, Federico Manuel Giorgi.

**Supervision:** Keng Boon Wee.

**Validation:** Paolo Pigini.

**Visualization:** Paolo Pigini.

**Writing – original draft:** Paolo Pigini.

**Writing – review & editing:** Paolo Pigini, Federico Manuel Giorgi, Keng Boon Wee.

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
