## [Decision Letter · Decision Letter 0]

14 Apr 2025

PCOMPBIOL-D-24-02008

Utargetome: a targetome prediction tool for modified U1-snRNAs to identify distal-target positions with improved selectivity

PLOS Computational Biology

Dear Dr. Wee,

Thank you for submitting your manuscript to PLOS Computational Biology. After careful consideration, we feel that it has merit but does not fully meet PLOS Computational Biology's publication criteria as it currently stands. Therefore, we invite you to submit a revised version of the manuscript that addresses the points raised during the review process.

Please submit your revised manuscript within 30 days Jun 14 2025 11:59PM. If you will need more time than this to complete your revisions, please reply to this message or contact the journal office at ploscompbiol@plos.org. Please include the following items when submitting your revised manuscript:

We look forward to receiving your revised manuscript.

Kind regards,

Bishoy Kamel

Guest Editor

PLOS Computational Biology

Shihua Zhang

Section Editor

PLOS Computational Biology

**Additional Editor Comments :**

Thanks for submitting the manuscript to PLoS Computional Biology, Two reviewers agreed to review the manuscript and have concluded it needs minor but critical changes. To point a few: The function of U5 and U6 snRNAs in base-pairing with the 5’ splice site subsequent to U1 should be elaborated, ensuring that their roles are either adequately recognized or explicitly refuted about modified U1s binding to distal sites. Furthermore, Figure 1 requires modification to offer a more thorough elucidation of the endogenous U1 structure, specifically highlighting the function of pseudouridines in U1 binding in addition to a clear indication of the numbering scheme. The notion of Minimum Annealed Bases (MAB) necessitates clarification, specifically about whether it encompasses solely Watson-Crick base pairs or also considers G-U wobble pairings, which naturally arise in 5’ splice site/U1 interactions. The descriptions for Figures 4 and 5 need improvements to clearly highlight the major findings and facilitate a direct comparison between the expected Utargetome.

Please make the recommended changes as the reviewers suggested and respond back with a point-by-by point rebuttal.

**Journal Requirements:**

3) Some material included in your submission may be copyrighted. According to PLOSu2019s copyright policy, authors who use figures or other material (e.g., graphics, clipart, maps) from another author or copyright holder must demonstrate or obtain permission to publish this material under the Creative Commons Attribution 4.0 International (CC BY 4.0) License used by PLOS journals. Please closely review the details of PLOSu2019s copyright requirements here: PLOS Licenses and Copyright. If you need to request permissions from a copyright holder, you may use PLOS's Copyright Content Permission form.

Potential Copyright Issues:

i) Figures 1A, 4D, and 4E. Please confirm whether you drew the images / clip-art within the figure panels by hand. If you did not draw the images, please provide (a) a link to the source of the images or icons and their license / terms of use; or (b) written permission from the copyright holder to publish the images or icons under our CC BY 4.0 license. Alternatively, you may replace the images with open source alternatives. See these open source resources you may use to replace images / clip-art:

ii) The following Figures contain a logo or branding: 1B, and 2. We are not permitted to publish this under our CC-BY 4.0 license, even with permission. We ask that you please remove or replace it.

4) Please amend your detailed Financial Disclosure statement. This is published with the article. It must therefore be completed in full sentences and contain the exact wording you wish to be published.

5) Please ensure that the funders and grant numbers match between the Financial Disclosure field and the Funding Information tab in your submission form. Note that the funders must be provided in the same order in both places as well. Currently, "Italian Ministry of University and Research, with the following programs: PON “Ricerca e Innovazione” 2014–2020; PRIN project 2022CEHEX8; PNRR program for HPC, Big Data, and Quantum Computing" are missing from the Funding Information tab.

6) Please provide a completed 'Competing Interests' statement, including any COIs declared by your co-authors. If you have no competing interests to declare, please state "The authors have declared that no competing interests exist." . 

**Reviewers' comments:**

Reviewer's Responses to Questions

Reviewer #1: This manuscript reports a new bioinformatics tool to predict the off-targets of mutant U1 snRNAs which are under evaluation for splice-correcting therapeutics, and this algorithm should improve the design of such modified U1s for high effectiveness and low off-target effects, mainly the latter. This algorithm accommodates exceptions of the target 5’ss and U1 base-pairing register, and its efficiency is supported by its ability to recognize most bona-fide human 5’ splice sites with the sequence of the endogenous U1. By analysing 54 modified U1s tested in other works, they found a range of predicted off-targets, and that targeting few nucleotides downstream of the canonical 5’ss (distal sites) shows the least predicted off-targets. This study should be useful to guide the assessment of the off-targets of modified U1s, to improve their design for therapeutic applications. My comments below are minor yet necessary for improving this manuscript:

1. The role of the U5 and U6 snRNAs in the base-pairing to different portions of the 5’ splice site after U1 should be better described as this is now too succinct. While the binding of modified U1s to the identified distal sites should not be followed by U5 and U6 at these sites, authors can ignore their contribution for the mode of action of modified U1s, yet this point needs to be addressed properly.

2. Figure 1 shows the endogenous U1 with its 5’ end. Since the two pseudouridines in the wild-type U1 contribute to the binding of 5’ splice sites, these modified nucleotides should be mentioned more extensively in the text, with 1-2 more citations.

3. Considering that the base-pairing between the 5’ end of U1 and the 5’ splice site is an RNA:RNA interaction, there are algorithms to estimate the free energy (deltaG) of binding, which often correlates with splicing efficiency. Likewise, the deltaG of modified U1s can also be estimated for off targets. Overall, authors would need to compare their Utargetome algorithm with the free-energy of binding.

4. We are also missing a more detailed definition of MAB as minimum annealed bases. Do MABs only correspond to Watson-Crick base pairs? Do MABs allow wobble base pairs (G-U) which also occur in 5’ splice site / U1 helices?

5. The data shown in Figure 4 and its in-text description in Results needs better explanations, and the main conclusions emphasized, to make sure the main point of the work is clear. How does the predicted Utargetome compare to experimentally derived off-targets reported in other works? Same notion applies to Figure 5.

Reviewer #2: The manuscript by Pigini and colleagues describes a bioinformatic investigation into the specificity of splicing changes induced by the expression of U1 snRNAs that are designed to rescue a splicing event that has been compromised by mutations in the sequence recognised by wild-type U1 snRNA. Using previous evidence for the variety of ways in which U1 snRNA can base-pair with 5'SS, they first look at possible targets in the transcriptome for wild-type U1 snRNA and then analyse the possible targets for 54 modified U1 snRNAs that have been shown elsewhere to restore splicing to defective sites. The goal of the work is to establish whether there is an optimum position near the target 5' splice site to which a rescuing U1 snRNA should be directed so as to minimize off-target effects.

The first part of the Results is devoted to establishing that the method is valid by showing that it correctly identifies sites at which the wild-type U1 snRNA might bind or act. The evidence given is that known 5'SS are identified (Fig. 3C, etc.), with U1 snRNA predicted to bind predominantly at the correct position (Fig. 3D). Most of the predicted sites of interaction (99.9%, from Figures 3A and B) were not known 5'SS, but no attempt was made to investigate whether they were in fact sites of base-pairing interactions. This could have been established by the use of existing data from psoralen cross-linking.

The second part of the results is based on an analysis of 54 modified U1 snRNAs that have been used previously for rescuing 5'SS mutations. After some analysis of the distribution of potential hits according to the particular sequence tested and the number of complementary base-pairs, the authors analysed RNAseq datasets to identify exons that might be affected by U1-36, which was designed to rescue a specific retinopathy. They identified 39 exons with fully complementary 5'SS that might be affected by U1-36. However, no data were shown to show that these exons had been affected by U1-36. More to the point, no use was made of the results obtained by analysing all the possible alignments and base-pairing configurations in the manner described earlier.

The final part of the results addresses the question about the optimal position, from the point of view of minimizing binding to non-target sites. Questions about the optimal site from the point of view of the mechanisms involved were not considered. Three modified U1 genes, shown to rescue splicing, were considered for each of two genes and shown to exhibit a wide range of potential off-target binding sites. The authors then analysed the potential 'targetome' for over thirty thousand U1 snRNAs that would be candidates for rescuing 839 5' splice site mutations, although the analysis was restricted to only perfectly complementary targets. The results suggested that U1 annealing from position +1 (to + 11) of the 5' splice site might be optimal. This result was taken further by demonstrating that the 54 modified U1 snRNAs studied earlier might have been more selective if the +1 design had been used.

The practicality of the approach illustrated in Figure 2 for assessing rescue U1s is not evident from the results, since two critical analyses looked only at perfectly complementary matches (Figures 4E and 5C). The other limitation is the failure to have examined experimental data to test whether there is a correlation between the predicted targetome and actual 5' splice sites affected by the rescue U1s or sites of contacts made in the transcriptome by the rescue U1s. These points should be discussed.

Minor points:

The manuscript would be improved by a diagram in Figure 1 that show very clearly what the numbering system means. I infer that, where there is fully complementary base-pairing, -3 or +1 means the position in the canonical splice site to which nucleotide 11 of U1 snRNA base-pairs. However, if, with a site that would normally by designated as -3 (the wild-type U1 alignment), U1 nucleotides 9, 10 and 11 did not base-pair, would the site in that alignment still be referred to as -3, or would it now be +1? Some illustrations of the various possibilities are needed, especially when considering bulges or loops.

**Have the authors made all data and (if applicable) computational code underlying the findings in their manuscript fully available?**

Reviewer #1: **No: **

Reviewer #2: Yes

PLOS authors have the option to publish the peer review history of their article (what does this mean? ). If published, this will include your full peer review and any attached files.

**Do you want your identity to be public for this peer review?** For information about this choice, including consent withdrawal, please see our Privacy Policy .

Reviewer #1: No

Reviewer #2: No

**Figure resubmission:**
---

## [Decision Letter · Decision Letter 1]

17 Sep 2025

Dear Dr. Wee,

We are pleased to inform you that your manuscript 'Utargetome: a targetome prediction tool for modified U1-snRNAs to identify distal-target positions with improved selectivity' has been provisionally accepted for publication in PLOS Computational Biology.

Best regards,

Bishoy Kamel

Guest Editor

PLOS Computational Biology

Shihua Zhang

Section Editor

PLOS Computational Biology

Reviewer #1:

Reviewer #2:

Reviewer's Responses to Questions

**Comments to the Authors:**

Reviewer #1: All ok now

Reviewer #2: The additions made by the authors to the manuscript have clarified some points and made it easier to understand others.

**Have the authors made all data and (if applicable) computational code underlying the findings in their manuscript fully available?**

Reviewer #1: Yes

Reviewer #2: Yes

PLOS authors have the option to publish the peer review history of their article (what does this mean? ). If published, this will include your full peer review and any attached files.

**Do you want your identity to be public for this peer review?** For information about this choice, including consent withdrawal, please see our Privacy Policy .

Reviewer #1: No

Reviewer #2: No

---

## [Editor Report · Acceptance letter]

PCOMPBIOL-D-24-02008R1

Utargetome: a targetome prediction tool for modified U1-snRNAs to identify distal-target positions with improved selectivity

Dear Dr Wee,

I am pleased to inform you that your manuscript has been formally accepted for publication in PLOS Computational Biology. Your manuscript is now with our production department and you will be notified of the publication date in due course.

With kind regards,

Zsofia Freund
